# Neonatal Exposure to Agonists and Antagonists of Sex Steroid Receptors Affects AMH and FSH Plasma Level and Their Receptors Expression in the Adult Pig Ovary

**DOI:** 10.3390/ani10010012

**Published:** 2019-12-19

**Authors:** Katarzyna Knapczyk-Stwora, Malgorzata Grzesiak, Patrycja Witek, Malgorzata Duda, Marek Koziorowski, Maria Slomczynska

**Affiliations:** 1Department of Endocrionology, Institute of Zoology and Biomedical Research, Faculty of Biology, Jagiellonian University, Gronostajowa 9, 30-387 Krakow, Poland; m.e.grzesiak@uj.edu.pl (M.G.); patrycja.witek@doctoral.uj.edu.pl (P.W.); maja.duda@uj.edu.pl (M.D.); maria.slomczynska@uj.edu.pl (M.S.); 2Department of Physiology and Reproduction of Animals, Institute of Biotechnology, University of Rzeszow, Werynia 502, 36-100 Kolbuszowa, Poland; mkozioro@ur.edu.pl

**Keywords:** AMH, endocrine active compounds, FSH, ovary, pig

## Abstract

**Simple Summary:**

The ovarian development and the establishment of ovarian reserve during fetal and/or neonatal life is critical for future reproductive success. Many environmental chemicals are known to negatively affect development and physiology of human and animal ovaries by interfering with endocrine systems, resulting in aberrant reproductive functions. The present study shows the long-term impact of neonatal exposure to agonists and antagonists of sex steroid receptors on AMH and FSH signalling in the ovary of adult pigs. Our findings suggest alteration in ovarian follicle recruitment from ovarian reserve arising from neonatal disruption of androgen/estrogen signalling induced by environmental endocrine active compounds. Everyday use of many endocrine disruptors is already prohibited after their harmful impacts on normal physiology have become known. Nevertheless, market introduction of new chemicals with potential deleterious influence on reproductive physiology has continued. Our outcomes confirm that a neonatal window plays an essential role in the physiological programming of ovarian function in adult pigs. The influence of environmental chemicals on this critical neonatal window needs to be investigated in order to gain a comprehensive view of deleterious interactions between endocrine disrupting chemicals and ovarian function.

**Abstract:**

In this study piglets were injected with testosterone propionate (TP, an androgen), flutamide (FLU, an antiandrogen), 4-*tert*-octylphenol (OP, an estrogenic compound), ICI 182,780 (ICI, an antiestrogen) or corn oil (controls) between postnatal days 1 and 10 (*N* = 5/group). Then plasma anti-Müllerian hormone (AMH) and follicle stimulating hormone (FSH) concentration and the expression of their receptors were examined in the adult pig ovary. TP and FLU decreased plasma AMH and FSH concentration. In preantral follicles, TP resulted in upregulation of AMHR2 and FSHR expression, but decreased AMH protein abundance. FLU upregulated AMHR2 expression, while OP increased *FSHR* mRNA. In small antral follicles, OP upregulated ACVR1 and BMPR1A expression, while FLU increased *BMPR1A* mRNA. FLU and ICI resulted in upregulation of AMHR2 expression. TP and FLU upregulated AMH expression, while it was downregulated in response to OP or ICI. Moreover, OP and ICI resulted in downregulation of FSHR expression, while FLU decreased FSHR protein abundance. In conclusion, neonatal exposure to either agonist or antagonist of androgen receptor affected AMH and FSH signalling systems in preantral follicles. In small antral follicles these systems were influenced by compounds with estrogenic, antiestrogenic, and antiandrogenic activity. Consequently, these hormonal agents may cause an accelerated recruitment of primordial follicles and affect the cycling recruitment of small antral follicles in pigs.

## 1. Introduction 

The major function of ovarian follicle is the creation of microenvironment for the growth and maturation of the oocyte, which in turn is capable of being fertilized. Ovarian follicle development through primordial, primary, and secondary stages, preantral-early antral follicle transition as well as oocyte release during ovulation is tightly regulated by an orchestrated action intra- and extra-follicular factors, including growth factors, gonadotropins, and steroids [1]. One of them is anti-Müllerian hormone (AMH), which is a dimeric protein and a member of the TGFβ (transforming growth factor β) family of growth- and differentiation-regulating factors. In the ovary of many mammalian species, including pigs, AMH is mainly expressed by granulosa cells of preantral and small antral follicles [2]. Noteworthy in pigs, AMH expression has been recently reported also in luteal cells [2].

AMH is thought to regulate the recruitment rate of primordial follicles into the expanding follicle pool in order to prevent exhaustion of the follicular reserve in ovaries [3]. Furthermore, it inhibits responsiveness of growing follicles to follicle-stimulating hormone (FSH) modulating cycling recruitment of small antral follicles [2,4]. Molecular mechanisms involved in AMH action requires binding to the transmembrane dimeric serine–threonine kinase receptors comprised of the AMH-specific type II receptor (AMHR2), and one of three type I receptors: BMPR1A, BMPR1B, or ACVR1. Upon binding to AMHR2, the type I receptor is recruited to form a hetero-oligomeric complex of type I and II receptors to initiate downstream signalling through activation of SMAD-1, SMAD-5, and SMAD-8 (for a review see [5]). 

In pigs, as in many mammalian species, folliculogenesis begin during fetal development, but the formation of the primordial follicle pool is completed around post-partum day 25 [6]. Thus both prenatal and neonatal periods are crucial during the establishment of the ovarian reserve and female reproductive potency, which is determined by the size of this reserve and its depletion rate. A growing body of evidence suggests that fetal and neonatal exposure to steroid receptor agonists and antagonists, including environmental agents (endocrine active chemicals, EACs) contributes to serious reproductive problems in adult life [7,8]. The EACs displaying androgenic/antiandrogenic and/or estrogenic/antiestrogenic activities have the ability to modulate hormone function by mimicking or blocking endogenous steroids signal transduction pathways [9]. Our previous results demonstrated that neonatal exposure to androgen and estrogen agonists or antagonists affected the number of primordial and primary follicles in piglets [10]. Recently, we have revealed that the exposure to the same compounds, i.e., testosterone propionate (TP; a synthetic androgen), flutamide (FLU; a nonsteroidal antiandrogen), 4-*tert*-octylphenol (OP; compound with estrogenic activity), and ICI 182,780 (ICI; a pure antiestrogen), during neonatal window affected GDF9 and BMP15 signalling in ovaries of adult pigs suggesting the acceleration of initial follicle recruitment and disruption of small antral follicles development [11]. Since AMH shares with BMP15 its type I receptor, BMPR1B, whose expression was altered by neonatal EACs treatment [11], it seems that disruption of steroid signalling may also affect AMH signalling in the ovary of adult pigs. Interestingly, we have also demonstrated the altered expression of FSH receptor (FSHR) in the neonatal porcine ovary following prenatal exposure to FLU pointing out the function of FSH in the promotion of early follicular development in pigs [12]. Based on these results, we hypothesized that androgen and estrogen availability during the neonatal window is essential for normal AMH and FSH plasma level as well as their cognate receptors expression ensuring proper ovarian function in adulthood. Therefore, the current study were directed toward characterizing the effects of neonatal exposure to TP, FLU, OP, and ICI on the AMH and FSH plasma level and their receptors expression in the adult pig ovary. To address this aim, the expression of ACVR1, BMPR1A, AMHR2, AMH, and FSHR in preantral and small antral ovarian follicles was assessed by real-time PCR, Western blot, and immunohistochemistry analyses as well as plasma AMH and FSH level by enzyme-linked immune sorbent assay.

## 2. Materials and Methods 

### 2.1. Design of Experiment and Sample Collection

Animal care and all experiments involving animals followed national guidelines and were approved by the Ethics Committee of the Jagiellonian University in Krakow, Poland (permit no. 150/2013, 122/2014, 123/2014, 187/2014 and 188/2014). The experimental design was as previously described [11]: twenty-five newborn female pigs (Large White × Polish Landrace) were included in the study. Piglets were separated randomly into five groups and injected subcutaneously with: (1) testosterone propionate (20 mg/kg body weight [bw]; *N* = 5; Sigma-Aldrich, St. Louis, MO, USA), (2) flutamide (50 mg/kg bw; *N* = 5; Sigma-Aldrich), (3) 4-*tert*-octylphenol (100 mg/kg bw; *N* = 5; Sigma-Aldrich), (4) ICI 182,780 (400 µg/kg bw; *N* = 5; Sigma-Aldrich), or (5) vehicle only (corn oil), which served as a control group (CTR, *N* = 5). Chemicals were administered daily for the period of postnatal day-1 through day-10. Doses of the chemicals were selected based on the information in the literature [13,14,15,16] and our previous experiments [10]. After weaning, the animals were kept until they reached sexual maturity. Following two estrous cycles of normal duration, 10- to 11-month old gilts were sacrificed on Days 8–11 after estrus at a local abattoir and ovaries were collected. Before slaughtering, blood from the jugular vein was collected and plasma was separated from blood samples by spinning at 2000× *g* for 10 min at 4 °C. Ovaries were kept in ice-cold phosphate buffered saline (PBS; pH 7.4, PAA The Cell Culture Company, Etobicoke, ON, Canada) supplemented with antibiotic/antimycotic solution (AAS 10 µl/mL; PAA The Cell Culture Company), and transferred to the laboratory within 2 hrs of collection. Small antral follicles (2–4 mm) were taken out from the ovary and either fixed in Bouin’s fixative for analysis by immunohistochemistry (IHC) or snap frozen in liquid nitrogen as two halves for protein and RNA preparation. 

To purchase preantral follicles (primordial, primary, and early secondary), we used a protocol involving enzymatic digestion [11]. Briefly, ovarian cortical tissue was rinsed in Dulbecco’s PBS medium (PAA The Cell Culture Company) and put into a tissue sectioner (Tissue Slicer Coronal, World Precision Instruments, Sarasota, FL, USA) to obtain uniform-size pieces of 1 × 1 × 1 mm. The enzymatic digestion of the tissue was performed in a digestion medium, consisting of 10 mL of PBS containing 0.08 mg/mL Liberase TH (Thermolysin High, Sigma-Aldrich, USA) and 0.2 mg/mL DNase (Sigma-Aldrich) for 120 min at 37 °C with gentle shaking. Then an equal volume of PBS medium at 4 °C supplemented with 10% fetal bovine serum (FBS) (Sigma-Aldrich) was added to terminate the enzymatic digestion. The digested cortex was filtered through nylon filters (Greiner Bio-One, GmbH, Germany)—first with a filter of pore size 70 µm, then with a filter of pore size 40 µm—and then spun down at 50× *g* for 10 min at 4 °C. The pellet was stored as snap frozen in liquid nitrogen and used later to prepare RNAs and protein lysates. In addition, a fragment of the cortex from each ovary was stored for IHC analysis after fixing in Bouin’s solution.

### 2.2. Hormone Assays 

The plasma AMH and FSH levels were determined using enzyme-linked immunosorbent assay (ELISA) with commercially available pig Muellerian-inhibiting factor ELISA kit (Wuhan EIAab Science Co., Ltd., Wuhan, China) and pig follitropin subunit beta ELISA kit (Wuhan EIAab Science Co., Ltd.), respectively, according to the manufacturer’s instructions. Assay sensitivity was 0.091 ng/mL for AMH and 0.39 mIU/ml for FSH. The intra-assay CV was ≤5.6% and 7.2% and the inter-assay CV was ≤7.8% and 10.1% for AMH and FSH, respectively. All analyses were performed in duplicate.

### 2.3. Quantitative Real-Time PCR

Total RNA from preantral follicles (*N* = 5, each group) and small antral follicles (*N* = 3, each animal) were isolated using TRI Reagent solution (Ambion, Austin, TX, USA). Total RNAs were reverse transcribed using a high-capacity cDNA reverse transcription kit (Applied Biosystems, Foster City, CA, USA), and real-time quantitative PCR of cDNAs was conducted using TaqMan Gene Expression Master Mix (Applied Biosystems) and porcine-specific TaqMan Gene Expression Assay (Applied Biosystems) for: *BMPR1A* (assay ID: Ss04248558_m1), *AMHR2* (assay ID: Ss04321772_m1), *AMH* (assay ID: Ss03383931_m1), and *FSHR* (assay ID: Ss03384581_u1) according to manufacturers’ protocol and run on StepOne™ Real-Time PCR System (Applied Biosystems). Primers for the analysis of *ACVR1* expression were designed based on the gene sequences in Ensembl database using Primer3 software (http://bioinfo.ut.ee/primer3/) and the assay was done using SYBR Green master mix (Applied Biosystems). Amplifications were performed with the StepOne™ Real-Time PCR System (Applied Biosystems) according to the recommended cycling conditions including melting curve analysis (ramp +0.5°C) to confirm the absence of primer dimmers. Primers used were as follows: for *ACVR1*; forward, 5′-CATCAGCTTAGCCAGAGAGGTT-3′; reverse, 5′-AGGTGGATTGCTTCGATTCTTA-3′; for *GAPDH*; forward, 5′-TGCTGTAGCCAAATTCATTGTC-3′; reverse, 5′-GATGACATCAAGAAGGTGGTGA-3′. Genomic DNA amplification contamination was checked by omitting reverse transcriptase during the RT step, non-template control was included in each run. All qPCR reactions were carried out in duplicates.

Relative mRNA quantification data were analyzed using the real-time PCR Miner algorithm [17]. The real-time PCR data obtained for *ACVR1*, *BMPR1A*, *AMHR2*, *AMH*, and *FSHR* were normalized to endogenous control, glyceraldehyde-3-phosphate dehydrogenase (*GAPDH*; assay ID: Ss03373286_u1).

### 2.4. Western Blotting

Protein extraction from preantral follicles (*N* = 5 per each group) and small antral follicles (*N* = 3 per each animal) and Western blot analysis was performed as previously described [18]. Equal amounts of protein (20 µg) were fractionated by 12% sodium dodecyl sulfate-polyacrylamide gel electrophoresis under reducing conditions [19] and transferred onto a PVDF membranes. Blocking of nonspecific binding sites on membranes was conducted using Tris-buffered saline (0.05 M Tris-HCl, pH 7.4) + 0.2% Tween20 (TBST) containing 5% (*v*/*v*) non-fat dry milk (1 h, room temperature with shaking). Then the overnight incubation (4 °C) with primary antibodies listed in Table 1 was performed. Next membranes were incubated with secondary anti-rabbit (BMPR1A, AMHR2, AMH, and FSHR) or anti-goat (ACVR1) antibodies linked to horseradish peroxidase (1:10,000; Jackson ImmunoResearch, Cambridge, UK) (1 h, room temperature). The sites of antibody-antigen binding were detected by chemiluminescence using Western Bright Quantum substrate (Advansta, Menlo Park, CA, USA) and visualized using ChemiDoc-It 410 Imaging System (UVP, Upland, CA, USA). Subsequently, to control the variable amounts of protein, each membrane was stripped and reprobed with monoclonal mouse anti-β-actin antibody (1:3000; Sigma-Aldrich) followed by horseradish peroxidase-conjugated anti-mouse IgG (1:10,000; Jackson ImmunoResearch). Bands were quantitated densitometrically using ImageJ software (National Institutes of Health, Bethesda, Maryland, USA). The Western blot data obtained for ACVR1, BMPR1A, AMHR2, AMH, and FSHR were normalized to its corresponding β-actin protein abundance. Each sample was analyzed in duplicate.

### 2.5. Immunohistochemistry

IHC was conducted as before [11]. Goat serum (5%, for BMPR1A, AMHR2, and FSHR) or horse serum (5%, for ACVR1) was used to block unspecific binding sites followed by an incubation with the primary antibodies listed in Table 1, overnight at 4 °C. Next respective biotinylated secondary antibodies (anti-rabbit or anti-goat IgGs; 1:300; 1.5 h; room temperature; Vector Laboratories, Burlingame, CA, USA) and avidin-biotinylated horseradish peroxidase complex (1:100; 40 min; room temperature; Dako, Glostrup, Denmark) were applied in succession. Finally, the antigen-antibody complex was detected upon incubation with 3,3′-diaminobenzidine (DAB, Sigma-Aldrich). Sections were counterstained with Hematoxylin QS (Vector Laboratories). Non-immune rabbit or goat IgG was used in place of primary antibody to ensure absence of non-specific staining. The stained sections were examined and photographed under a Nikon Eclipse Ni-U microscope using Nikon Digital DS-Fi1-U3 camera (Nikon, Tokyo, Japan) and Nikon NIS-ELEMENT Image Software. 

### 2.6. Statistical Analysis

Data, expressed as mean ± SEM, were analyzed using Statistica v.13.1 software (StatSoft, Inc., Tulsa, OK, USA). We used nonparametric Mann-Whitney U-test to assess the significance of differences between the control (CTR, untreated) and TP-, FLU-, OP- or ICI-treated groups. Results were considered statistically significant at *p* < 0.05. Each individual gilt was used as the experimental unit. For small antral follicles data, two-way mixed ANOVA was performed in each examined group to test the relative magnitudes of the intra- and inter-subjects differences. Since there were no significant differences between follicles within individual gilt, the average of the three follicles from individual gilt was used making the gilt an experimental unit.

## 3. Results 

### 3.1. Plasma AMH and FSH Concentrations

Plasma AMH concentration decreased following neonatal TP (*p* < 0.01) and FLU (*p* < 0.05) administration when compared with the control group (Figure 1a). Plasma FSH level was diminished (*p* < 0.05) in both TP- and FLU-treated groups than in the control group (Figure 1b). OP or ICI treatment did not alter the plasma levels of AMH and FSH (Figure 1a,b, respectively).

### 3.2. Expression of mRNA and Protein for ACVR1, BMPR1A, AMHR2, AMH, and FSHR in Preantral Follicles 

Effect of agonists and antagonists of sex steroid receptors on mRNA expression and protein abundance of ACVR1, BMPR1A, AMHR2, AMH, and FSHR in the preantral follicles was examined using quantitative real-time PCR (Figure 2a–e, respectively) and Western blot analyses (Figure 2a’–e’, respectively). In both control and EACs-treated pigs, examined proteins were detected in preantral follicles and shown in Figure 2f. The approximately observed molecular weights were: 56 kDa (ACVR1), 60 kDa (BMPR1A), 55 kDa (AMHR2), 61 (AMH), and 78 kDa (FSHR).

In the preantral follicle population, both ACVR1 and BMPR1A mRNA expression and protein abundance were not affected by EACs treatment (Figure 2a,a’ and Figure 2b,b’, respectively). *AMHR2* mRNA expression were greater for the TP- and FLU-treated groups (*p* < 0.001 and *p* < 0.01, respectively) relative to the control group (Figure 2c). Treatments with TP and FLU also led to the increased abundance of AMHR2 protein (*p* < 0.05) (Figure 2c’). EACs-treatment did not alter *AMH* mRNA levels, although protein levels for AMH were reduced in the TP-treated group (*p* < 0.05) when compared to the control group (Figure 2d’).TP- and OP-treated groups showed marked increase in *FSHR* mRNA levels (*p* <0.01) (Figure 2e). However, level of FSHR protein were higher in the TP-treated group relative to the control group (*p* < 0.05) (Figure 2e’).

### 3.3. Expression of mRNA and Protein for ACVR1, BMPR1A, AMHR2, AMH, and FSHR in Small Antral Follicles 

Effect of agonists and antagonists of sex steroid receptors on mRNA expression and protein abundance of ACVR1, BMPR1A, AMHR2, AMH, and FSHR in small antral follicles was examined using quantitative real-time PCR (Figure 3a–e, respectively) and Western blot analyses (Figure 3a’–e’, respectively). Analyzed proteins showed bands as approximately 56 kDa (ACVR1), 60 kDa (BMPR1A), 55 kDa (AMHR2), 61 (AMH), and 78 kDa (FSHR) in small antral follicles from control and EACs-treated pigs as shown in Figure 3f. 

The expression of *ACVR1* mRNA increased both in the FLU- and OP-treated groups (*p* < 0.01 and *p* < 0.05, respectively) in comparison to the control group (Figure 3a). However, ACVR1 protein abundance was higher only in the OP-treated group (*p* < 0.05) when compared to the control one (Figure 3a’). *BMPR1A* mRNA and protein levels markedly increased (*p* < 0.05) in the OP-treated group (Figure 3b,b’, respectively). *AMHR2* mRNA levels were markedly greater both in the FLU- and ICI-treated groups (*p* < 0.01 and *p* < 0.05, respectively) when compared to the control group (Figure 3c). Treatments with FLU and ICI also led to significantly higher (*p* < 0.05) AMHR2 protein levels than the control group (Figure 3c’). AMH mRNA and protein levels were markedly increased (*p* < 0.05) in the TP- and FLU-treated groups (Figure 3d,d’). On the contrary, administration of OP and ICI resulted in a significantly lower mRNA expression (*p* < 0.01 and *p* < 0.05, respectively) and protein abundance (*p* < 0.05) of AMH compared to the control group (Figure 3d,d’). The expression of *FSHR* mRNA markedly decreased both in the OP- and ICI-treated groups (*p* < 0.01 and *p* < 0.001, respectively, Figure 3e), while FSHR protein abundance was lower in the FLU-, OP-, and ICI-treated groups (*p* < 0.05, Figure 3e’) compared to the control group.

### 3.4. Immunolocalization of ACVR1, BMPR1A, AMHR2, and FSHR in Ovarian Follicles

The positive ACVR1 (Figure 4), BMPR1A (Figure 5), AMHR2 (Figure 6), and FSHR (Figure 7) staining was found in all examined sections. In preantral follicles of all groups, ACVR1, BMPR1A, AMHR2, and FSHR proteins were detected in oocytes and granulosa cells (Figure 4a–e, Figure 5a–e, Figure 6a–e, and Figure 7a–e, respectively). In small antral follicles of test groups, ACVR1, BMPR1A, and AMHR2 proteins were detected in both granulosa and theca cells (Figure 4a’–e’, Figure 5a’–e’, and Figure 6a’–e’), while FSHR protein was found in granulosa cells (Figure 7a’–e’). Negative controls were presented as insets in Figure 4a’, Figure 5a’, Figure 6a’ and Figure 7a’.

## 4. Discussion

Recently, we have demonstrated that neonatal exposure to EACs displaying androgenic/antiandrogenic and/or estrogenic/antiestrogenic activities affected oocyte-derived growth factors signalling system in the adult porcine ovary, indicating that neonatal period is vital for programming of ovarian function in pigs. As a consequence, the acceleration of initial follicle recruitment and disturbed antral follicle development have been proposed [11]. To extend these lines of research, the present study investigated the effects of those EACs administration during the neonatal period on AMH and FSH signalling in ovarian follicles of the adult pig. Since AMH-mediated inhibition of primordial follicles recruitment and modulation of the sensitivity of growing follicles to FSH has been described [2,3], we suggest that the long-term effects of EACs on ovarian function additionally may be driven by altered AMH and FSH signalling. 

The growing follicles develop from a reserve of primordial follicles that contains all of the oocytes potentially available for fertilization throughout the female lifespan [20]. We have previously shown that neonatal exposure to TP and FLU interrupted the earliest stages of folliculogenesis in 11-day-old pigs and may cause the acceleration of follicle initial recruitment [10]. Findings from AMH null mice revealed that AMH inhibited the initial recruitment of primordial follicles into the growing pool [21]. The results presented herein show the decrease in plasma AMH concentration in mature pigs that were neonatally exposed to either TP or FLU, while it was not affected by OP and ICI treatment. On the other hand, AMH protein abundance in preantral follicle homogenates was decreased only upon neonatal TP administration. These results further confirm the acceleration of primordial follicle recruitment in response to neonatal androgen or antiandrogen treatment herein employing AMH signalling.

It is known that AMH binds to its corresponding receptor type II (AMHR2), and the type I receptors (ACVR1, BMPR1A, or BMPR1B) are subsequently recruited to form a heterodimeric complex [5]. Although ACVR1 and BMPR1A expression were not affected by EACs treatment herein, recently we have shown the upregulation of BMPR1B expression in preantral follicle population upon neonatal TP and ICI administration [11]. In the current study, neonatal administration of both TP and FLU led to the upregulation of AMHR2 mRNA expression and protein abundance in preantral follicles of adult pigs. Similarly, in adult rats prenatally exposed to androgen the AMHR2 expression was higher, whereas serum concentration of AMH was lower [22]. These results indicate that the elevated level of AMHR2 expression in preantral follicles might be the ovarian response to diminished plasma AMH concentration in adult pigs that were neonatally treated with either TP or FLU. 

Besides the inhibitory role of AMH during initial recruitment, it may modify preantral follicular growth by reducing their responsiveness to FSH [23]. FSH may speed up the initiation of the primordial follicles growth [21]. We have previously demonstrated the downregulation of FSHR in preantral follicles of neonatal pigs exposed prenatally to FLU, which confirms a key role of androgens in porcine folliculogenesis at the early stages [12]. In the current study, similar to the present findings for AMH, neonatal administration of either TP or FLU resulted in lower plasma FSH concentration. Moreover, FSHR expression at mRNA and protein level was increased in preantral follicles from TP-exposed pigs at the neonatal life. Androgens have also been shown to exert a stimulatory effect on growth of preantral follicles by increasing FSH responsiveness via up-regulation of FSHR expression [24]. Considering our results, lower plasma AMH and FSH level as well as higher AMHR2 and FSHR expression in preantral follicles, which result from neonatal androgen excess or deficiency, may promote the initial recruitment of follicles in pigs. Interestingly, similar effect of androgens and antiandrogens is consistent with our previous study examining the influence of neonatal exposure to EACs on the expression of oocyte-derived growth factors and their cognate receptors in ovarian follicles in adult pigs. We proposed that both androgen excess and deficiency during the neonatal window is critical and may lead to the similar long-term effects [11]. Since altered epigenetic regulation of gene expression by EACs has been proposed as a mechanism for long-term effects on ovarian physiology [25], this mechanism of similar response to TP and FLU cannot be excluded. 

The role of androgens in the early stage of folliculogenesis and during the preantral to early antral transition is considered to be more significant than the role of estrogens [26]. Our earlier findings demonstrated that prenatal FLU treatment alters the expression of the selected set of members of the TGFβ superfamily and cognate receptors, including AMH, BMPR1B, and AMHR2, indicating the importance of androgens during early follicle development in pigs [27]. However, the cycling recruitment and follicle development beyond preantral stage is FSH-dependent and characterized by an increase in follicular estrogen production. In addition to direct action on gonadotropin release from the anterior pituitary gland, estrogens are obligatory for normal antral follicle development [28]. It is known that AMH decreases the responsiveness of small antral follicles to FSH and inhibited FSH-stimulated estradiol production [2]. Hence, it is not surprising that the results of the present study showed the prominent effect of compounds with estrogenic and antiestrogenic activity in small antral follicles. ACVR1 and BMPR1A expression were upregulated upon neonatal OP administration, while AMHR2 expression was upregulated in response to neonatal ICI and FLU treatment. In addition, in small antral follicles, neonatal OP or ICI treatment resulted in decreased AMH expression, while it was increased in response to neonatal TP and FLU treatment. Abbott et al. [29] summarized the effect of estrogen excess or deficiency on ovarian function in adulthood, suggesting that fetal exposure to chemicals derived from environment or diet binding to estrogen receptors may cause reprogramming of ovarian function in adulthood. In addition, we have previously demonstrated that neonatal exposure to antiandrogenic and estrogenic compounds altered GDF9 and BMP15 signalling networks within the antral follicle and may disturb the follicular steroidogenesis and readiness to further growth [11]. Results herein further confirm the long-term effects of EACs with antiandrogenic, estrogenic, and antiestrogenic activity on small antral follicle development in pigs as the impaired AMH signalling through type I and type II receptors is possible. However, further studies are required to elucidate the impact on intracellular signalling pathways.

Liu et al. [30] demonstrated that high levels of testosterone inhibit follicle development via FSH signalling suppression in cultured granulosa cells. In the present study, FSHR expression in small antral follicles decreased in response to neonatal FLU, OP, or ICI treatment suggesting the impact of these compounds on cycling recruitment. Interestingly, plasma FSH levels were diminished in mature pigs that were neonatally treated with TP and FLU, but not in those exposed to OP or ICI treatment. However, it seems that either lower FSHR expression or plasma FSH level disturbed the cycling recruitment of small antral follicles and estradiol production when androgen and estrogen action is disrupted during the neonatal window. Moreover, altered plasma FSH level in pigs that were exposed to TP or FLU in the neonatal life suggests the long-term effect on the disruption of the pituitary-ovarian axis. 

In the current study, in both control and EACs-treated ovaries, ACVR1, BMPR1A, AMHR2, and FSHR were localized in the oocytes and granulosa cells of preantral follicles. Moreover, small antral follicles obtained from all examined groups ACVR1, BMPR1A, AMHR2 were observed in both granulosa and theca cells, while FSHR was present in granulosa cells. Our data indicate the sites of AMH and FSH action within ovarian follicles and confirm previous findings showing the presence of BMPR1A, AMHR2, and FSHR in the porcine ovary [12,18,27,31]. However, to our knowledge, results of the present study provide the first evidence for ACVR1 localization in porcine ovaries.

## 5. Conclusions

In conclusion, lower plasma AMH and FSH level suggests acceleration of primordial follicle recruitment and disturbed cycling recruitment of small antral follicle in pigs when steroid action is altered during the neonatal window by androgen excess or deficiency. Moreover, neonatal exposure to EACs affected AMH and FSH signalling systems in preantral follicles, while in small antral follicles these systems were affected in response to compounds with estrogenic, antiestrogenic, and antiandrogenic activity. Considering our results, we confirm a critical contribution of the neonatal window to functional programming of ovaries during adulthood in pigs [11], and further indicate that the long-term effects of steroids excess or deficiency on follicle initial and cycling recruitment may be, at least in part, mediated by altered AMH and FSH signalling. Since EACs are commonly used in the modern living environment, the long-term effects of steroid excess or deficiency EACs should be extensively studied.

## Figures and Tables

**Figure 1 animals-10-00012-f001:**
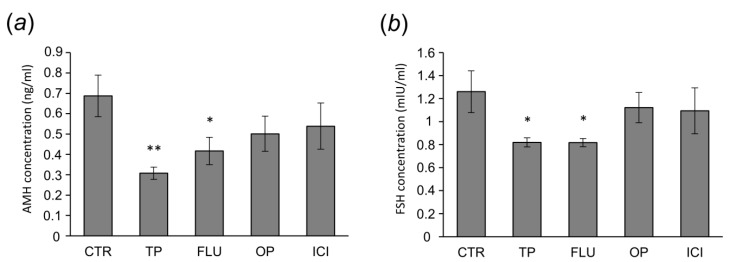
Plasma anti-Müllerian hormone (AMH) (**a**) and follicle stimulating hormone (FSH) (**b**) concentrations in control (CTR) and testosterone propionate—(TP), flutamide—(FLU), 4-*tert*-octyl-phenol—(OP), and ICI 182,780-treated (ICI) adult pigs. Data are expressed as the mean ± standard error of the mean (*N* = 5). Asterisks denote significant differences between control and treated animals, * *p* < 0.05, ** *p* < 0.01, Mann-Whitney U test.

**Figure 2 animals-10-00012-f002:**
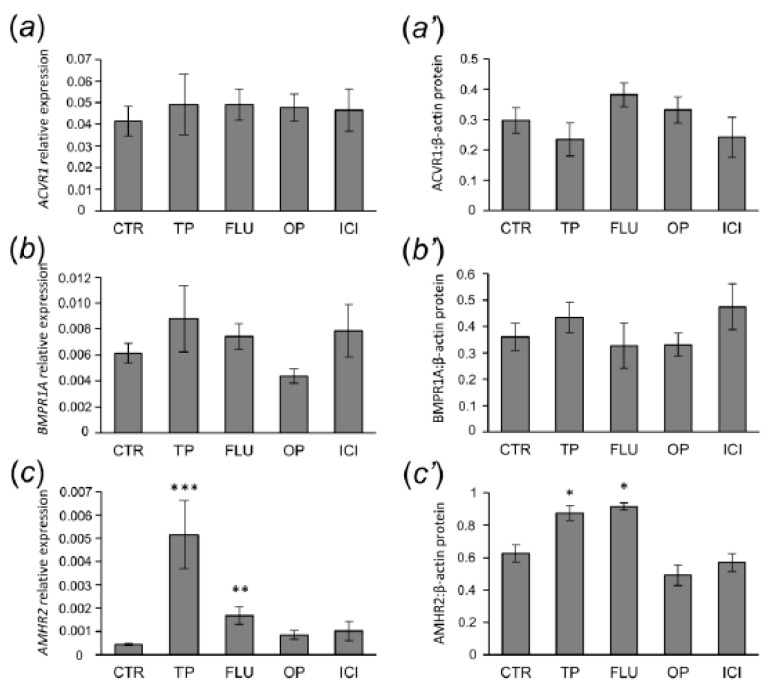
ACVR1 (**a**,**a’**), BMPR1A (**b**,**b’**), AMHR2 (**c**,**c’**), AMH (**d**,**d’**), and FSHR (**e**,**e’**) mRNA expression and protein abundance in primordial, primary, and early secondary follicles obtained from control (CTR), testosterone propionate—(TP), flutamide—(FLU), 4-*tert*-octylphenol—(OP), and ICI 182,780-treated (ICI) pigs. The mRNA expression was determined using quantitative real-time PCR and presented relative to glyceraldehyde-3-phosphate dehydrogenase as mean ± standard error of the mean (a–e). Relative abundance of protein was evaluated densitometrically and expressed as the ratio relative to β-actin abundance (mean ± standard error of the mean; (a’–e’)). (**f**) The fragment of membrane with bands corresponding to predicted molecular weights are shown. Asterisks denote significant differences between the control and treated animals, * *p* < 0.05, ** *p* < 0.01, *** *p* < 0.001, Mann-Whitney U test.

**Figure 3 animals-10-00012-f003:**
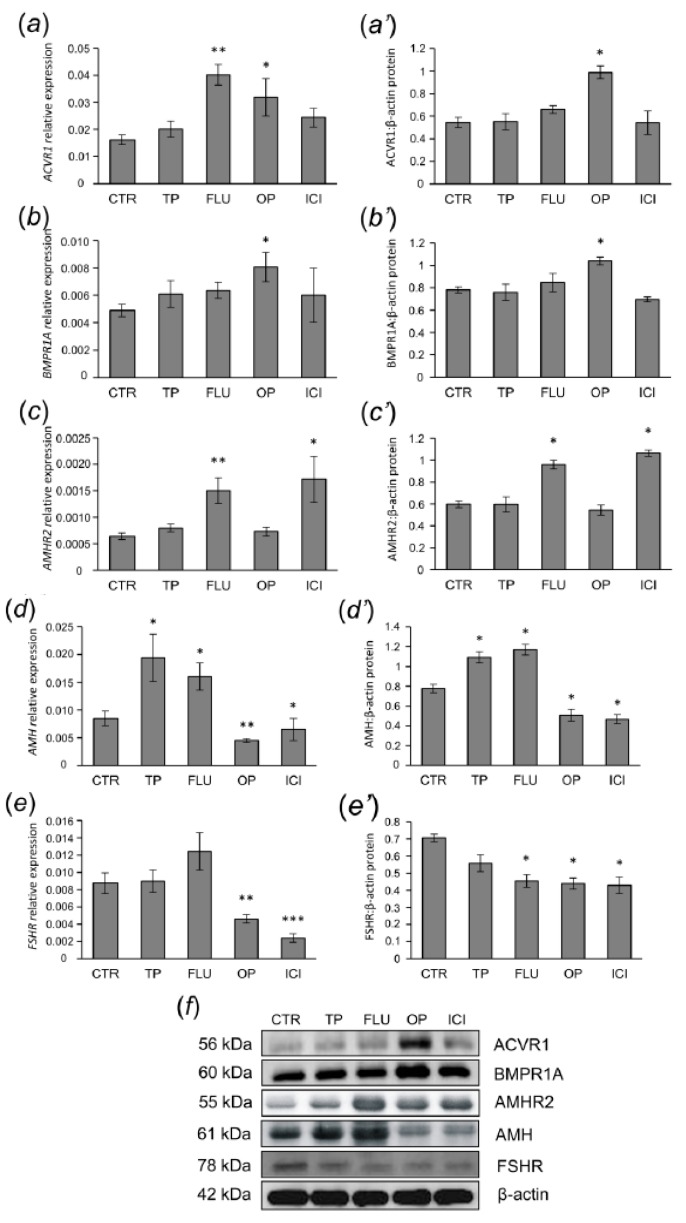
ACVR1 (**a**,**a’**), BMPR1A (**b**,**b’**), AMHR2 (**c**,**c’**), AMH (**d**,**d’**) and FSHR (**e**,**e’**) mRNA expression and protein abundance in small antral follicles obtained from control (CTR), testosterone propionate—(TP), flutamide—(FLU), 4-*tert*-octylphenol—(OP), and ICI 182,780-treated (ICI) pigs. The mRNA expression was determined using quantitative real-time PCR and presented relative to glyceraldehyde-3-phosphate dehydrogenase as mean ± standard error of the mean (a–e). Relative abundance of protein was evaluated densitometrically and expressed as the ratio relative to β-actin abundance (mean ± standard error of the mean; (a’–e’)). (**f**) The fragment of membrane with bands corresponding to predicted molecular weights are shown. Asterisks denote significant differences between control and treated animals, * *p* < 0.05, ** *p* < 0.01, *** *p* < 0.001, Mann-Whitney U test.

**Figure 4 animals-10-00012-f004:**
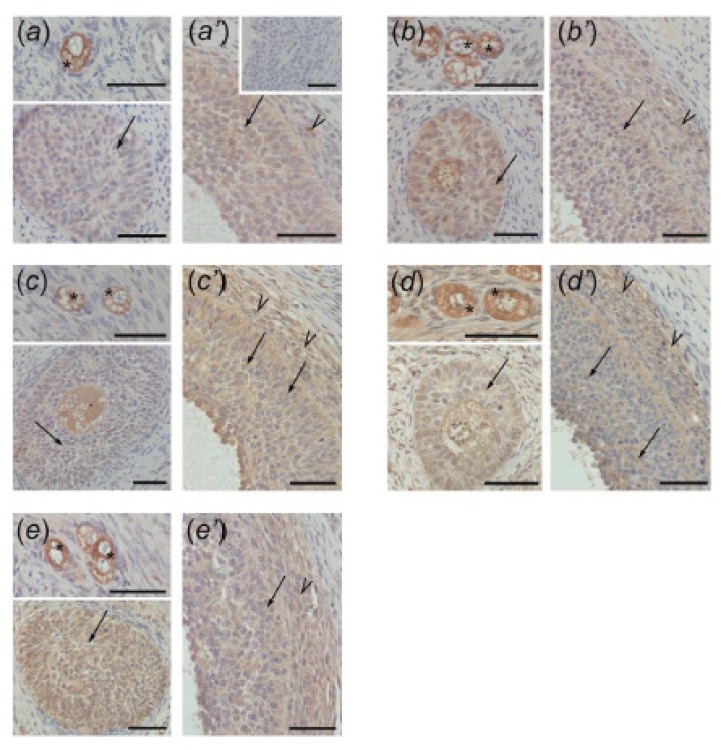
ACVR1 immunostaining in the preantral (**a–e**) and small antral (**a’–e’**) follicles obtained from control gilts (**a**,**a’**) and treated with testosterone propionate (**b**,**b’**), flutamide (**c**,**c’**), 4-*tert*-octylphenol (**d**,**d’**), and ICI 182,780 (**e**,**e’**). Immunopositivity was detected in oocytes (asterisks) and granulosa cells (arrows) of preantral follicles (a–e) as well as in both granulosa (arrows) and theca (arrowheads) cells of small antral follicles (a’–e’). Hematoxylin QS was used for counterstaining sections. Control sections showed no positive staining ((a’) inset). Bars = 50 µm.

**Figure 5 animals-10-00012-f005:**
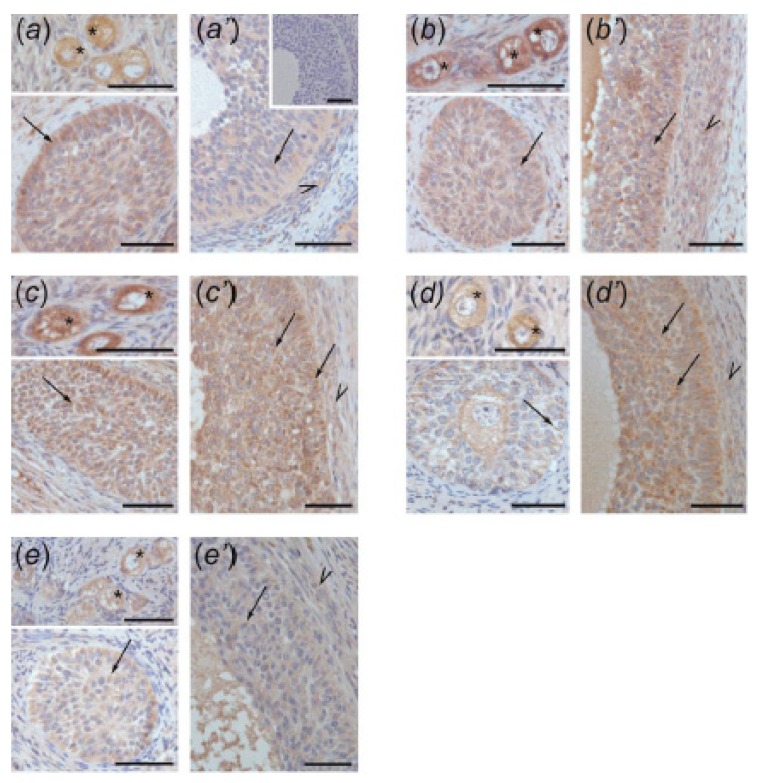
BMPR1A immunostaining in the preantral (**a–e**) and small antral (**a’–e’**) follicles obtained from control gilts (**a**,**a’**) and treated with testosterone propionate (**b**,**b’**), flutamide (**c**,**c’**), 4-*tert*-octylphenol (**d**,**d’**), and ICI 182,780 (**e**,**e’**). Immunopositivity was detected in oocytes (asterisks) and granulosa cells (arrows) of preantral follicles (a–e) as well as in both granulosa (arrows) and theca (arrowheads) cells of small antral follicles (a’–e’). Hematoxylin QS was used for counterstaining sections. Control sections showed no positive staining ((a’) inset). Bars = 50 µm.

**Figure 6 animals-10-00012-f006:**
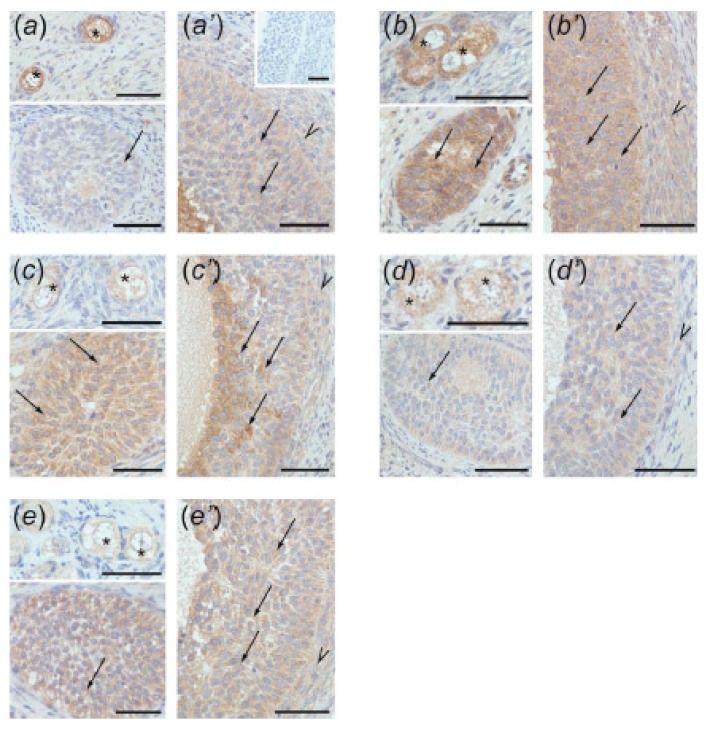
AMHR2 immunostaining in the preantral (**a–e**) and small antral (**a’–e’**) follicles obtained from control gilts (**a**,**a’**) and treated with testosterone propionate (**b**,**b’**), flutamide (**c**,**c’**), 4-*tert*-octylphenol (**d**,**d’**), and ICI 182,780 (**e**,**e’**). Immunopositivity was detected in oocytes (asterisks) and granulosa cells (arrows) of preantral follicles (a–e) as well as in both granulosa (arrows) and theca (arrowheads) cells of small antral follicles (a’–e’). Hematoxylin QS was used for counterstaining sections. Control sections showed no positive staining ((a’) inset). Bars = 50 µm.

**Figure 7 animals-10-00012-f007:**
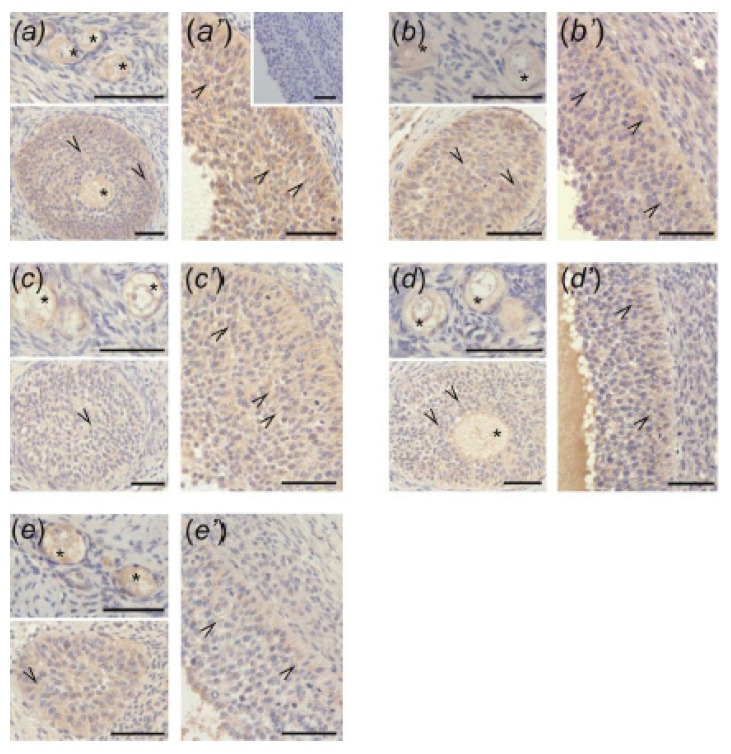
FSHR immunostaining in the preantral (**a–e**) and small antral (**a’–e’**) follicles obtained from control gilts (**a**,**a’**) and treated with testosterone propionate (**b**,**b’**), flutamide (**c**,**c’**), 4-*tert*-octylphenol (**d**,**d’**), and ICI 182,780 (**e**,**e’**). Immunopositivity was detected in oocytes (asterisks) and granulosa cells (arrowheads) of preantral follicles (a–e) as well as granulosa cells (arrowheads) of small antral follicles (a’–e’). Hematoxylin QS was used for counterstaining sections. Control sections showed no positive staining ((a’) inset). Bars = 50 µm.

**Table 1 animals-10-00012-t001:** List of primary antibodies used for immunoblotting and IHC.

Antibody	Dilution Used for		Host	Type	Supplier
	WB	IHC			
Anti-ACVR1	1:1000	1:200	Goat	Polyclonal	Acris Antibodies GmbH, Herford, Germany (AP22507PU-N)
Anti-BMPR1A	1:1000	1:100	Rabbit	Polyclonal	Kindly provided by Prof. C. H. Heldin (Ludwig Institute for Cancer Research Ltd., Uppsala, Sweden)
Anti-AMHR2	1:1000	1:100	Rabbit	Polyclonal	LifeSpan BioSciences Inc., Seattle, WA, USA (LS-B11943)
Anti-AMH	1:1000	-	Rabbit	Polyclonal	Acris Antibodies GmbH, Herford, Germany (TA336233)
Anti-FSHR	1:1000	1:100	Rabbit	Polyclonal	Bioss Antibodies, Woburn, MA, USA (bs-0895R)

ACVR1 = activin A receptor type 1; BMPR1A = bone morphogenetic protein receptor type 1A; AMHR2 = anti-Mullerian hormone type II receptor; AMH = anti-Mullerian hormone; FSHR = follicle-stimulating hormone receptor; IHC = immunohistochemisty; WB = Western blot.

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
