# Peer review of "Neonatal Exposure to Agonists and Antagonists of Sex Steroid Receptors Affects AMH and FSH Plasma Level and Their Receptors Expression in the Adult Pig Ovary"

_animals, 2019, doi:10.3390/ani10010012_

Round 1

Reviewer 1 Report

General Comments: 

This is an interesting study built on several other studies from this group showing the influence of androgens on folliculogenesis in gilts.  My biggest concern is that the experimental unit is gilt (by definition, the experimental unit is the smallest unit to which the treatment is applied).  When multiple follicles are collected from a treated subject such as has occurred in these studies, follicle must be nested within gilt and there is no description of that.  If follicle is used as the experimental unit, it artificially increases the denominator degrees of freedom, reduces the error sum of squares and inflates the F value.  While I know the authors have published several papers from these tissues and while I believe this is interesting and relevant work, the experimental unit must be gilt, this must be stated and the statistical analysis must be described better,

Specific Comments:

Title is mis-leading.  The investigators did not look at AMH or FSH signaling at all.

Line 86:  This is not the aim.  The investigators did not look at signaling at all.  For FSH, the signaling pathway would be cAMP, for AMH, the signaling pathway would be SMADs.

Line 86:  Please state a hypothesis.  The scientific method requires a hypothesis to be tested.  The investigators performed statistics, they presumably tested a hypothesis.  Placing a hypothesis at the end of the Introduction aids the reader in understanding the focus of the study.

Line 96:  Treated subcutaneously is vague.  Presumably this was an injection.

Line 108:  Days 8 – 11 of the estrous cycle represent a long span in the status of the follicles.  There was no discussion of including day of the estrous cycle in the statistical model.

Line 109 – 111:  What is the experimental unit?  Technically gilt is the experimental unit because gilt is the smallest unit to which treatment was applied.  Therefore, all small follicles within a gilt represent a n of 1 and in the statistical model follicles should be nested within gilt.  Again, there is no description of this.

Line 112:  Primordial, primary, and secondary follicles were combined within gilt for protein extraction?  By mass, secondary follicles would dwarf the contribution of primordial and primary follicles.  This informs us very little about what is going on at the individual preantral stages and most likely represents secondary follicles more than anything else.

Line 193 -196:  Biggest concern is still the experimental unit.  Gilt not follicle is the experimental unit.  Was it analyzed this way?

Author Response

Thank you very much for the revision of our paper. All your comments were very useful and we followed them while rewriting the manuscript.

Comment 1. My biggest concern is that the experimental unit is gilt (by definition, the experimental unit is the smallest unit to which the treatment is applied).  When multiple follicles are collected from a treated subject such as has occurred in these studies, follicle must be nested within gilt and there is no description of that.  If follicle is used as the experimental unit, it artificially increases the denominator degrees of freedom, reduces the error sum of squares and inflates the F value. While I know the authors have published several papers from these tissues and while I believe this is interesting and relevant work, the experimental unit must be gilt, this must be stated and the statistical analysis must be described better.

Response 1. Thank you for your crucial remark. We do agree that the experimental unit should be a gilt, as it is for analysis of plasma AMH and FSH concentration as well as expression of mRNA and protein for ACVR1, BMPR1A, AMHR2, AMH, and FSHR in preantral follicles. However, we have previously noticed that antral follicles may differ from each other in many aspects. It is known that single ovarian follicle is a separate ovarian unit, that may response in the different way. In our experiment three small antral follicles were obtained from the same animal. So, if we have pooled the small antral follicles mRNA and protein obtained from the same animal and have analysed them as a single sample, there is a possibility to obtain a negligible differences. Therefore in our project we have decided to analysed the follicles as separate units. However, we thank you for this remark and in the future study we will take it into consideration. 

Comment 2. Title is mis-leading. The investigators did not look at AMH or FSH signaling at all.

Response 2. Title of the Manuscript has been modified into: “Neonatal Exposure to Agonists and Antagonists of Sex Steroid Receptors Affects AMH and FSH Plasma Level and Their Receptors Expression in the Adult Pig Ovary”. 

Comment 3. Line 86:  This is not the aim.  The investigators did not look at signaling at all.  For FSH, the signaling pathway would be cAMP, for AMH, the signaling pathway would be SMADs.

Response 3. The following changes have been made in the revised MS to address this comment: "Therefore, the current study were directed toward characterizing the effects of neonatal exposure to TP, FLU, OP, and ICI on the AMH and FSH plasma level and their receptors expression in the adult pig ovary” (lines 87-90 in revised MS).

Comment 4. Line 86:  Please state a hypothesis. The scientific method requires a hypothesis to be tested.  The investigators performed statistics, they presumably tested a hypothesis.  Placing a hypothesis at the end of the Introduction aids the reader in understanding the focus of the study.

Response 4. The hypothesis has been stated in lines 85-87 of revised MS.

Comment 5. Line 96: Treated subcutaneously is vague. Presumably this was an injection.

Response 5. It was corrected in the revised MS (line 100 in revised MS).

Comment 6. Line 108: Days 8 – 11 of the estrous cycle represent a long span in the status of the follicles. There was no discussion of including day of the estrous cycle in the statistical model.

Response 6. Most of samples derived from days 9-10 of estrous cycle. However follicles from few animals were obtained on days 8 or 11 of estrous cycle. Our recently published paper revealed that the radioimmunological analysis of steroids concentration in plasma within each group revealed similar results (Witek et al. Domest Anim Endocrinol. 2020; 70:106381. doi: 10.1016/j.domaniend.2019.07.009). Moreover, there were no significant differences between samples within each examined group referring to examined proteins.           

Comment 7. Line 109 – 111:  What is the experimental unit?  Technically gilt is the experimental unit because gilt is the smallest unit to which treatment was applied.  Therefore, all small follicles within a gilt represent a n of 1 and in the statistical model follicles should be nested within gilt.  Again, there is no description of this.

Response 7. Please see the Response 1

Comment 8. Line 112:  Primordial, primary, and secondary follicles were combined within gilt for protein extraction?  By mass, secondary follicles would dwarf the contribution of primordial and primary follicles.  This informs us very little about what is going on at the individual preantral stages and most likely represents secondary follicles more than anything else.

Response 8. As it is stated in Materials and Method section in lines 124-126, after tissue digestion, the filtration by nylon filters with pore size of 70 µm and then 40 µm was applied. Thus, there were mostly follicles with a diameter between 30 and 40 µm, i.e. primordial and primary ones as well as only early secondary follicles. (lines 125-127 in revised MS)

Comment 9. Line 193 -196:  Biggest concern is still the experimental unit.  Gilt not follicle is the experimental unit.  Was it analyzed this way?

Response 9. Please see the Response 1.

Reviewer 2 Report

Neonatal exposure to agonists and antagonists of sex steroid receptors affects AMH and FSH  signalling in the adult pig ovary

Authors: Knapczyk-Stwora K, et al.  

The authos findings suggest that alteration in ovarian follicle recruitment from  ovarian reserve arising from neonatal disruption of androgen/estrogen signalling induced by environmental endocrine active compounds is essential for programming of ovarian function in adult pigs and the risk of future reproductive disorders.

The area is very interesting, however its relevance for this process is still unclear. This paper adds a further level of complexity in that it outlines a neonatal disruption of androgen/estrogen signalling involvement in this process.

In general, the methods used are appropriate, and appear to have been carefully carried out. My specific comments regarding the manuscript are as follows:

Minor

It is not clear, whether the authors pooled all the samples of particular group for Western blotting analysis or the present pictures represent only the best gels? Information concerning the secondary antibody used for WB should be presented. General: gene names are to be written in italics if referring to mRNA. It is not clear, whether RT-PCR and WB were performed in duplicates or triplicates. It should be described in Materials and Methods. It is normal practice to quantify WB using a minimum 4 identical blots and compare the expression with a housekeeping internal control. Please provide details of the controls.

Author Response

The authors express sincere thanks to the reviewer for all her/his comments and suggestions which were very helpful and contributed to the quality of the paper. Following your specific comments we have revised our manuscript.

Comment 1. It is not clear, whether the authors pooled all the samples of particular group for Western blotting analysis or the present pictures represent only the best gels?

Response 1. The samples were not pooled, the pictures show the representative gel from each analysis.

Comment 2. Information concerning the secondary antibody used for WB should be presented.

Response 2. This information was previously presented in lines 169-170: Next membranes were incubated with secondary anti-rabbit (BMPR1A, AMHR2, AMH, and FSHR) or anti-goat (ACVR1) antibodies linked to horseradish peroxidase (1:10000; Jackson ImmunoResearch, Cambridge, UK) (1 hour, room temperature) (lines 169-171 in revised Manuscript)

Comment 3. General: gene names are to be written in italics if referring to mRNA.

Response 3. It has been corrected.

Comment 4. It is not clear, whether RT-PCR and WB were performed in duplicates or triplicates. It should be described in Materials and Methods. It is normal practice to quantify WB using a minimum 4 identical blots and compare the expression with a housekeeping internal control. Please provide details of the controls.

Response 4. The information referring to RT-PCR was previously presented in lines 154-155 (lines 155-156 in revised MS). The information referring to WB has been added into revised version of MS (lines 180-181 in revised MS). We have quantify WB results using 5 identical blots and β-actin as internal control.

Round 2

Reviewer 1 Report

Overall, the authors have adequately addressed the majority of my concerns.

I understand the authors concern about the physiological differences among antral follicles within a gilt.  This does not, however, change the determination of the experimental unit.  This can be viewed as similar to examining a dominant follicle and a subordinate follicle in a wave in a cow.  The physiology of the two follicles is absolutely different.  If you have 6 treated cows and 6 control cows you could examine the follicular fluid steroid concentration of the two types of follicles within a cow.  You would use a repeated measure model with cow as the subject. In this way the error sum of squares would be divided by 10 (12 cows - 1 df for treatment -1 df for the error term).  This would be the denominator for the F-statistic.  If you did not correct for the 2 types of follicles within a cow the error sum of squares would be divided by 22 (24 follicles - 1 df for treatment - 1 df of freedom for the error term).  This would results in a much smaller denominator for the F-statistic and artificially inflate the F-statistic.

Even if the authors feel that the three follicles within a gilt are different they must either come up with a way that classifies those differences and model it within gilt  or in the absence of that they must simply nest the three follicles within gilt in a repeated measures model (thereby taking the average of the three follicles within a gilt and making gilt the experimental unit).  I am sorry.  My laboratory struggles with these complexity of experimental design for folliculogenesis within cows, sheep and pigs as well, but it does not change the proper way of doing the research.

Author Response

The authors express their sincere thanks to the Reviewer for his comment. Following your suggestion and statistician consultation of our data we have re-analyzed them. Each individual gilt was used as the experimental unit in all analysis. For small antral follicles data, two-way mixed ANOVA was performed in each examined group (CTR, TP-, FLU-, OP-, and ICI-treated) to test the relative magnitudes of the intra- and inter-subjects differences. Since there were no significant differences between follicles within individual gilt, the average of the three follicles from individual gilt was used making the gilt an experimental unit. We have also improved the description of statistical analysis (Lines 242-246 in revised MS).